# Reliability and Validity of the Educational Stress Scale for Adolescents (ESSA) in a Sample of Greek Students

**DOI:** 10.3390/children10020292

**Published:** 2023-02-02

**Authors:** Evangelia Moustaka, Flora Bacopoulou, Kyriaki Manousou, Christina Kanaka-Gantenbein, George P. Chrousos, Christina Darviri

**Affiliations:** 1Postgraduate Course of Science of Stress and Health Promotion, School of Medicine, National and Kapodistrian University of Athens, 11527 Athens, Greece; 2University Research Institute of Maternal and Child Health Hospital and Precision Medicine and UNESCO Chair on Adolescent Health Care, Medical School, National and Kapodistrian University of Athens, Aghia Sophia Children’s Hospital, 11527 Athens, Greece; 3First Department of Pediatrics, Medical School, National and Kapodistrian University of Athens, Aghia Sophia Children’s Thivon & Papadiamantopoulou St., 11527 Athens, Greece; 4School of Medicine, University of Athens, Soranou Ephessiou Str. 4, 11527 Athens, Greece

**Keywords:** ESSA, validation, educational stress, Greek adolescents

## Abstract

This research outlines the initial validation of a new instrument to quantify academic stress, the Educational Stress Scale for Adolescents (ESSA). A total of 399 students (61.9% females, 38.1% males), with a mean age of 16.3 years, participated in the research protocol. Cronbach’s α for the total 16-item ESSA scale was 0.878, suggesting good reliability. Cronbach’s α for each one of the five components were statistically positively significant. The Greek version of the Educational Stress Scale for Adolescents (ESSA) can be utilized as a valid tool to measure the perceived educational stress in adolescents.

## 1. Introduction

Educational (or academic) stress can be understood as the mental discomfort that results from an expected frustration over an academic failure [1]. Furthermore, in school conditions, during the educational process, students may experience excessive pressure to succeed, achieve high grades, or ensure their entry into higher education [2]. Excessive stress during this stage could lead to an increased prevalence of mental health problems, including anxiety and depressive feelings, which could eventually have an adverse effect on the outcomes achieved [3].

For students, the educational period is a long and complex cognitive process requiring them to muster both mental and physical strength, emotional stability, psychological balance, goal orientation, and ability to overcome stress-induced barriers, particularly during examination periods [4]. Inevitably, the difficulties educational stress gives rise to may affect several other sectors in an individual’s life; indeed, stress may gradually be-come generalized [4].

Regarding the Greek experience and, more specifically, the dimensions of educational stress in Greece, what emerges from the relevant literature is that there is currently an insufficient number of studies on the subject. Thus, there are no systematic research studies measuring educational/school-related stress and its cognitive, mental, or emotional repercussions in the life of adolescents. However, there are some validated instruments measuring certain aspects of educational stress, one of which is the Adolescent Stress Questionnaire, which comprises such scales as school performance, interaction with teachers, school attendance, and school/leisure conflict [5]. Another example is the widely used Test Anxiety Inventory, which is mainly focused on deficiencies arising from examination-induced stress and anxiety [6].

As the relevant Greek literature makes clear, there are no validated and cross-culturally adapted questionnaires on school-related stress that clearly measure the multifactorial nature of educational stress. School education and entry into higher educational institutions constitute a crucial stage in a Greek adolescents’ life [6]. This societally determined notion, which is of great importance to an individual’s personal, social, and economic development, has clearly influenced, in fact determined, the orientation of the Greek educational system at different levels [7]. As discussed above, it is important for the adolescent research community to study the risk factors that contribute to educational stress and also the protective factors for the mental health of this population [8]. The aim of this study is the validation of the Educational Stress Scale (ESSA) in Greek adolescents.

## 2. Materials and Methods

### 2.1. Translation Procedure

The questionnaire’s validation was based on the process of its translation into Greek and its cross-cultural adaptation. After contacting the author responsible for the questionnaire and obtaining his/her written consent for its validation in Greek adolescents (January 2020), the original questionnaire was translated (forward translation) by two bilingual translators; subsequently, the translated questionnaire was back translated into English, the questionnaire’s original language (backward translation). Then, a pilot test was conducted with the target population, who were provided with the questionnaire. Finally, the procedure of translation and cross-cultural adaptation was duly evaluated and completed.

### 2.2. Participants and Procedures

A convenience sample of 399 students (61.9% female; 38.1% male), with a mean age of 16.3 years, participated in the study, which took place in Athens, Greece, between November 2021 and April 2022. The study was conducted after the research protocol was approved by the Committee of Research Ethics and Deontology of the School of Medicine of the National and Kapodistrian University of Athens (Ref. 93776 8/11/2021). The questionnaires were administered to the adolescent students after their parent/caregivers’ written consent was obtained.

### 2.3. Ethical Considerations

The protocol of this research was approved by the ethics committee of the Medical School of the National and Kapodistrian University of Athens and was in accordance with the Declaration of Helsinki (2013). For the completion of the printed questionnaire, volunteers who participated in the study signed a form of information, consent, and confidentiality.

### 2.4. Measures

For the purposes of our research, participants were first provided with a demographics questionnaire, including questions about their gender, age, and school year, as well as their parents’ educational level and work status. The questionnaire’s demographic factors also included questions about the students’ study habits, the number of hours they devoted to the Internet or leisure, as well as questions about their overall way of life and habits, as adapted from Sun et al. (2011) [8].

#### 2.4.1. Educational Stress Scale for Adolescents (ESSA)

This questionnaire under validation includes 16 items/questions aimed at assessing educational stress. Each question included a 5-point Likert-type scale (ranging from 1 = strongly disagree to 5 = strongly agree). The participants’ responses to the questions added up to a total score; moreover, 5 different factors resulted from the statistical analysis of their responses to the following: ‘Pressure from study’; ‘Work-load’; ‘Worry about grades’; ‘Self-expectation’, and ‘De-spondency’. Higher scores indicate greater stress. Published in English by Sun et al. (2011), the questionnaire has good internal consistency, with Cronbach’s α = 0.82 for the total scale and α = 0.79, α = 0.73, α = 0.69, α = 0.65, and α = 0.64 for the five factors, separately [8].

#### 2.4.2. Adolescent Stress Questionnaire—ASQ

Adapted to Greek by Darviri et al. (2014), the ASQ measures various stress-inducing events and situations in the life of adolescents using a 5-point Likert-type scale (ranging from 1 = no stress to 5 = a lot of stress). The internal consistency was Cronbach’s α value (range: 0.70–0.86) and general value: 0.96 [5].

#### 2.4.3. Rosenberg Self-Esteem Scale

This questionnaire measures self-esteem by employing a 4-point Likert-type scale (arranging from 1 = strongly agree to 4 = strongly disagree) with high reliability of α = 0.809. Higher scores in the scale indicate higher self-esteem levels [9].

#### 2.4.4. State-Trait Anxiety Inventory (STAI)

This questionnaire uses two subscales to measure state and trait anxiety. Cronbach’s α was 0.93 for the state anxiety scale and 0.92 for the trait anxiety subscale. Higher scores in the scale indicate greater stress [10].

### 2.5. Data Analysis

For the categorical variables, data are presented as N (%) and for the quantitative variables as median (IQR) and mean (SD). An exploratory factor analysis (EFA) was conducted for the 16 items of the ESSA. The Kaiser–Meyer–Olkin (KMO) measure and Bartlett’s test of sphericity were used to evaluate the adequacy of the sample. Direct oblique rotation was used and eigenvalues near to 1 showed the appropriate number of factors. Items with factor loadings greater than 0.3 were assigned to the factors. Cronbach’s alpha was used to evaluate the internal consistency. Spearman’s rho coefficient was used to correlate the quantitative variables, due to the skewed distribution of the quantitative variables. Statistical analyses were performed using SPSS v.26 for Windows, and the significance level for all analyses was 0.05.

## 3. Results

The research sample’s main demographic characteristics are presented in Table 1. Overall, 399 students (61.9% female) participated in the research, with a mean (SD) age of 16.3 (1.4) years.

Most of the students responded that, over the past year, they had spent over 2 h per day on assigned homework (80.2%) and had attended private classes (87.5%). In addition, 51.1% of the students reported having classes during weekends or holidays. As far as their school performance was concerned, they had received high or very high grades (65.4%).

Regarding the 30-day period prior to participating in the research, most of the respondents reported that they enjoyed a good health status (73.7%), engaged in physical activity (77.4%), used the Internet (99.7%), and played electronic games (54.6%).

### 3.1. Table Analysis (Scales Examined)

The IQR and the mean (SD) for the ESSA examined are displayed in Table 2. More specifically, for the entire ESSA score, the mean (SD) was 47.7 (10.0); for the ‘Pressure from study’, ‘Workload’, ‘Worry about grades’, ‘Self-expectation’, and ‘Despondency’ subscales, the values were 12.0 (3.0), 9.9 (2.5), 8.4 (2.5), 9.5 (2.7), and 8.0 (2.3), respectively.

### 3.2. ESSA Questionnaire—Principal Component Analysis

The factor loadings, resulting from the factor component analysis (FCA) of the ESSA Questionnaire’s 16 items, are presented in Table 3.

The following five components were extracted: ‘Self-expectation’, ‘Workload’, ‘Despondency’, ‘Pressure from study’, and ‘Worry about grades’. These components account for 36.3%, 10.9%, 7.1%, 6.1% and 5.9% of the fluctuation, respectively.

Table 3 shows the EFA results for the 16 items on the ESSA questionnaire, with direct oblique rotation. For examining the adequacy of the sample, the Kaiser–Meyer–Olkin (KMO) test and Barlett’s Sphericity Test were used. The KMO value was 0.883, and the significance of Bartlett’s test of sphericity was p < 0.001. The Scree plot, as a result of the EFA, indicated that five factors were extracted with an eigenvalue near to 1 (Figure 1). The 16 items explained the 66.27% of the total variance.

Cronbach’s α for the total 16-item ESSA range was 0.878, thus, suggesting good reliability (Table 4). Cronbach’s α for each one of the five components was 0.735, 0.768, 0.546, 0.685, and 0.621, respectively. All interitem correlations between the five components were statistically positively significant (Table 5, Spearman’s ρ: 0.324–0.627). Table 6 describes the correlations between the ESSA and its subscales with the other tools that were used in the study. There were statistically significant positive correlations between the ESSA subscales and those of the STAI and ASQ questionnaires (Table 6). The strongest correlations (ρ > 0.6) were observed between the scales of ‘Self-expectation’ and ‘STAI-trait’ (ρ = 0.668; p < 0.001), ‘Self-expectation’ and ‘STAI-state’ (ρ = 0.627; p < 0.001), ‘Pressure from study’ and ‘ASQ-Stress of Home Life’ (ρ = 0.617; p < 0.001), ‘Pressure from study’ and ‘ASQ-Stress of School Performance’ (ρ = 0.694; p < 0.001), ESSA total score and ‘STAI-trait’ (ρ = 0.678; p < 0.001), ESSA total score and ‘STAI-state’ (ρ = 0.618; p < 0.001), ESSA total score and ‘ASQ-Stress of Home Life’ (ρ = 0.646; p < 0.001), ESSA total score and ‘ASQ-Stress of School Performance (ρ = 0.709; p < 0.001), and ESSA total score and ‘ASQ-Stress of Future Uncertainty’ (ρ = 0.617; p < 0.001).

Statistically significant negative correlations were observed between the ESSA subscales and the ‘Self-esteem’ scale (ρ range: from −0.270 for the ‘Workload’ scale to −0.574 for the ‘Despondency’ scale).

## 4. Discussion

The ESSA questionnaire on adolescent educational stress was validated and cross-culturally adapted in this study. Furthermore, an exploratory factor analysis was conducted, and specific variables of the other questionnaires were examined. In the validation of the educational stress assessment instrument, 399 students took part, the majority of whom came from urban centres (89%). More specifically, based on the exploratory factor analysis (EFA) of the questionnaire’s 16 items, five components were extracted, as was the case with the original questionnaire [8], and the relevant variables were named: ‘Self-expectation’, ‘Workload’, ‘Despondency’, ‘Pressure from study’, and ‘Worry about grades’. These components account for 36.3%, 10.9%, 7.1%, 6.1%, and 5.9% of the fluctuation, respectively.

According to the statistical analysis conducted, Cronbach’s α reliability coefficient for the entire 16-item ESSA rate was 0.878, thus suggesting good reliability, as was also the case with the original questionnaire [8]. Furthermore, all of the interitem correlations between the five components were statistically positively significant (Spearman’s ρ range: 0.324–0.627).

Continuing with the correlations obtained from the analysis of the educational stress questionnaire and the remaining scales, a notable finding was the strong positive correlation between the ESSA scale “pressure from study” and the stress of home life scale (home life stress) from the ASQ Adolescent Stress Questionnaire. This particular finding is consistent with the literature where stress and conflict in the family environment influence and are influenced by the stress of studying [4]. Furthermore, a strong positive correlation was found between the ESSA scale “pressure from study” and the stress of school performance (ASQ-Stress of School Performance’ (rho = 0.694,·p < 0.001)). In addition, this finding of research is confirmed by the literature where it is known that the school performance of students is affected by the pressure of reading and studying [4].

Finally, statistically significant negative correlations were found between the ESSA subscales ‘workload’ and ‘despondency’ with the self-esteem scale. Adolescents who exhibited low levels of self-esteem scored higher on the educational stress scale, which is consistent with the existing literature and research hypotheses [11].

As with other similar research studies, this validation is not without some limitations. We used a ‘convenience sample’, in other words, an ‘opportunity sample’, i.e., lower and senior high school students who probably had some specific characteristics, which somewhat limited the generalizability of the findings. Moreover, the sample largely came from urban centres, as it was not possible to use additional samples from the provinces of Greece. This precluded taking into account potential geographical factors, which might influence educational stress variables.

In conclusion, the Greek version of the Educational Stress Scale for Adolescents generally has good psychometric properties, and it could safely be applied as a measure in future studies in Greece.

## Figures and Tables

**Figure 1 children-10-00292-f001:**
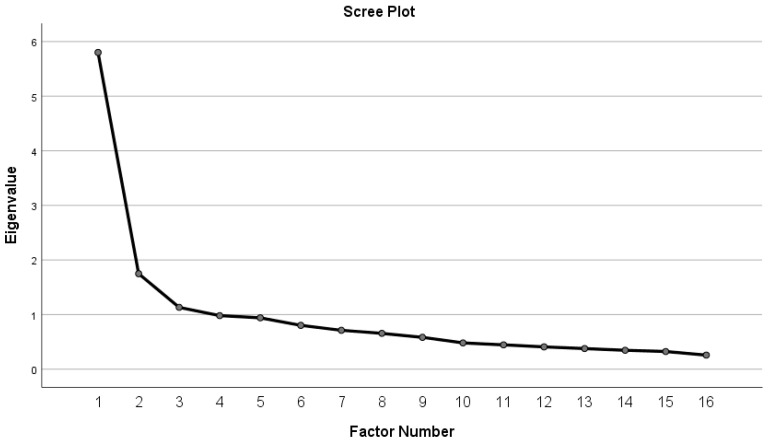
Depicts the Screen Plot, as a result of the EFA.

**Table 1 children-10-00292-t001:** Samples’ Sociodemographic characteristics (N = 399).

Gender N (%)	
- Males	152 (38.1)
- Females	247 (61.9)
Age (years)	
- Median (IQR)	17.0 (1.0)
- Mean (SD)	16.3 (1.4)
Class N (%)	
- Seventh grade (Gymnasium)	12 (3.0)
- Eighth grade (Gymnasium)	30 (7.5)
- Ninth grade (Gymnasium)	38 (9.5)
- Tenth grade (Lyceum)	94 (23.6)
- Eleventh grade (Lyceum)	115 (28.8)
- Twelfth grade (Lyceum)	110 (27.6)
Place of residence N (%)	
- Athens	355 (89.0)
- Other	44 (11.0)
Siblings N (%)	
- Yes	324 (81.2)
- No	75 (18.8)
Parents’ Marital Status N (%)	
- Married	291 (72.9)
- Divorced/Separated	100 (25.1)
- Death of one or both parents	8 (2.0)
Parents’ Educational status N (%)	
- Until Upper Secondary School (Lyceum)	127 (31.8)
- Bachelor	207 (51.9)
- MSc/PhD	65 (16.3)
Parents’ work status N (%)	
- Unskilled	22 (5.5)
- Semi-skilled	28 (7.0)
- Skilled	349 (87.5)
Time spent on schoolwork, daily N (%)	
- Less than 2 h	79 (19.8)
- 2–3 h	182 (45.6)
- More than 3 h	138 (34.6)
Private tutoring during the last year N (%)	
- Yes	349 (87.5)
- No	50 (12.5)
Class during weekend or holidays N (%)	
- Yes	204 (51.1)
- No	195 (48.9)
Grades during last year N (%)	
- Low	13 (3.3)
- Medium	125 (31.3)
- High	194 (48.6)
- Very high	67 (16.8)
Health state during the last 30 days N (%)	
- Bad	9 (2.3)
- Average	96 (24.1)
- Good	294 (73.7)
Physical exercise during the last 30 days N (%)	
- Yes	309 (77.4)
- No	90 (22.6)
Use of internet during the last 30 days N (%)	
- Yes	398 (99.7)
- No	1 (0.3)
Electronic games during the last 30 days N (%)	
- Yes	214 (53.6)
- No	185 (46.4)

**Table 2 children-10-00292-t002:** Samples’ study’s measurements (N = 399).

Scales and Subscales Scores	Median (IQR) Mean (SD)
ESSA-total score	49.0 (15.0) 47.7 (10.0)
ESSA-pressure from study	12.0 (4.0) 12.0 (3.0)
ESSA-workload	10.0 (4.0) 9.9 (2.5)
ESSA-worry about grades	8.0 (4.0) 8.4 (2.5)
ESSA-self-expectation	10.0 (5.0) 9.5 (2.7)
ESSA-despondency	8.0 (4.0) 8.0 (2.3)

**Table 3 children-10-00292-t003:** Rotated factor loadings: Exploratory Factor analysis (EFA) results regarding the ESSA questionnaire’s 16 items (N = 399).

	Component				
	Self-Expectation Stress	Workload	Despondency	Pressure from Study	Worry about Grades
14. I feel stressed when I do not live up to my own standards	0.716				
15. When I fail to live up to my own expectations, I feel I am not good enough	0.751				
16. I usually cannot sleep because of worry when I cannot meet the goals I set for myself	0.662				
2. I feel that there is too much school work		0.824			
3. I feel there is too much homework		0.791			
7. I feel that there are too many tests/exams in the school		0.600			
1. I am very dissatisfied with my academic grades			0.530		
12. I always lack confidence with my academic scores			0.636		
13. It is very difficult for me to concentrate during classes			0.346		
4. Future education and employment bring me a lot of academic pressure				0.600	
5. My parents care about my academic grades too much which brings me a lot of pressure				0.339	
6. I feel a lot of pressure in my daily studying				0.728	
11. There is too much competition among classmates which brings me a lot of academic pressure				0.487	
8. Academic grade is very important to my future and even can determine my whole life					0.419
9. I feel that I have disappointed my parents when my test/exam results are poor					0.733
10. I feel that I have disappointed my teacher when my test/exam results are not ideal					0.378
Eigenvalues	5.800	1.749	1.132	0.981	0.941
% of Variance	36.251	10.933	7.074	6.130	5.880

**Table 4 children-10-00292-t004:** Range, mean, standard deviation, correlations (item-total correlation), and Cronbach’s α coefficients for the ESSA questionnaire scales.

Items	Range	Mean (SD)	Item-Total Correlations	Alpha of Scale
ESSA Total	16–90	47.73 (10.04)		0.878
Self-expectation	3.0–15.0	9.48 (2.68)		0.735
14. I feel stressed when I do not live up to my own standards			0.582	
15. When I fail to live up to my own expectations, I feel I am not good enough			0.600	
16. I usually cannot sleep because of worry when I cannot meet the goals I set for myself			0.500	
Workload	3.0–15.0	9.90 (2.51)		0.768
2. I feel that there is too much schoolwork			0.694	
3. I feel there is too much homework			0.601	
7. I feel that there are too many tests/exams in the school			0.514	
Despondency	3.0–15.0	8 (2.26)		0.546
1. I am very dissatisfied with my academic grades			0.354	
12. I always lack confidence with my academic scores			0.382	
13. It is very difficult for me to concentrate during classes			0.345	
Pressure from study	4.0–20.0	11.99 (1.96)		0.685
4. Future education and employment bring me a lot of academic pressure			0.536	
5. My parents care about my academic grades too much that brings me a lot of pressure			0.346	
6. I feel a lot of pressure in my daily studying			0.597	
11. There is too much competition among classmates that brings me a lot of academic pressure			0.416	
Worry about grades	3.0–15.0	8.38 (2.47)		0.621
8. Academic grade is very important to my future and even can determine my whole life			0.294	
9. I feel that I have disappointed my parents when my test/exam results are poor			0.553	
10. I feel that I have disappointed my teacher when my test/exam results are not ideal			0.464	

**Table 5 children-10-00292-t005:** Correlations (Spearman’s rho) between ESSA subscales (N = 399).

	Self-Expectation	Workload	Despondency	Pressure from Study	Worry about Grades
Self-expectation	1.000				
Workload	0.420	1.000			
Despondency	0.526	0.324	1.000		
Pressure from study	0.627	0.596	0.564	1.000	
Worry about grades	0.487	0.339	0.548	0.573	1.000

**Table 6 children-10-00292-t006:** Correlations (Spearman rho) between ESSA and STAI, self-esteem and ASQ subscales.

		ESSA- Pressure from Study	ESSA- Workload	ESSA- Worry about Grades	ESSA- Self-expectation	ESSA- Despondency	ESSA-Total Score
STAI-Trait	Spearman rho	0.536	0.364	0.508	0.668	0.563	0.678
	p-value	<0.001	<0.001	<0.001	<0.001	<0.001	<0.001
STAI-State	Spearman rho	0.524	0.337	0.417	0.627	0.478	0.618
	p-value	<0.001	<0.001	<0.001	<0.001	<0.001	<0.001
Self-esteem	Spearman rho	−0.431	−0.270	−0.412	−0.572	−0.574	−0.573
	p-value	<0.001	<0.001	<0.001	<0.001	<0.001	<0.001
ASQ-Stress of Home Life	Spearman rho	0.617	0.336	0.473	0.503	0.579	0.646
	p-value	<0.001	<0.001	<0.001	<0.001	<0.001	<0.001
ASQ-Stress of School Performance	Spearman rho	0.694	0.544	0.424	0.546	0.534	0.709
	p-value	<0.001	<0.001	<0.001	<0.001	<0.001	<0.001
ASQ-Stress of School Attendance	Spearman rho	0.450	0.367	0.251	0.282	0.387	0.446
	p-value	<0.001	<0.001	<0.001	<0.001	<0.001	<0.001
ASQ-Stress of Romantic Relationships	Spearman rho	0.357	0.147	0.271	0.232	0.392	0.347
	p-value	<0.001	0.003	<0.001	<0.001	<0.001	<0.001
ASQ-Stress of Peer Pressure	Spearman rho	0.463	0.234	0.349	0.311	0.486	0.463
	p-value	<0.001	<0.001	<0.001	<0.001	<0.001	<0.001
ASQ-Stress of Teacher Interaction	Spearman rho	0.447	0.284	0.357	0.308	0.484	0.472
	p-value	<0.001	<0.001	<0.001	<0.001	<0.001	<0.001
ASQ-Stress of Future Uncertainty	Spearman rho	0.591	0.393	0.367	0.593	0.453	0.617
	p-value	<0.001	<0.001	<0.001	<0.001	<0.001	<0.001
ASQ-Stress of School/Leisure Conflict	Spearman rho	0.584	0.430	0.307	0.458	0.416	0.567
	p-value	<0.001	<0.001	<0.001	<0.001	<0.001	<0.001
ASQ-Stress of Financial Pressure	Spearman rho	0.481	0.270	0.279	0.377	0.429	0.470
	p-value	<0.001	<0.001	<0.001	<0.001	<0.001	<0.001
ASQ-Stress of Emerging Adult Responsibility	Spearman rho	0.381	0.261	0.226	0.260	0.290	0.360
	p-value	<0.001	<0.001	<0.001	<0.001	<0.001	<0.001

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
