# Peer review of "Reliability and Validity of the Educational Stress Scale for Adolescents (ESSA) in a Sample of Greek Students"

_children, 2023, doi:10.3390/children10020292_

Round 1

Reviewer 1 Report

Thank you for the opportunity to review this paper. This is a simple but useful paper that allows for the implementation of ESSA on Greek speakers. The paper is well written and follows the traditional narrative and logic of assessment tool adaptations to other languages. I have no major concerns with this paper.

Here are some minor suggestions that should be addressed.

There are many hyphenated words that should not be hyphenated, please correct this.

I suggest presenting the rotated factor plots along with scree plots even as supplementary. Factor loading are often a bit difficult to interpret without the visual aids of the rotated plots.

I am aware that the percent of variance is presented at the bottom of Table 3 and mentioned in lines 161-162, however I wonder if the amount of variance explained per factor is comparable to the original tool and to adaptation to other languages.

In summary I would endorse the publication of this tool provided these suggestions are considered.

Reviewer 2 Report

The article is of interest. It is important to enable measuring instruments, so this work is relevant. The translation process is correct. 

However, I find the following aspects that should be corrected:

Method

I miss a final section in method, data analysis, where the analyses to be performed are described.

It would be convenient to put the Greek version used for the STAI (line 113) and the Rosenberg Self-Esteem Scale (line 109).

The data in Table 1, being sample descriptors, should perhaps be placed in the participants section, under method.

Results

Principal component analysis is not a factor analysis. I would suggest redoing the analyses with an exploratory factor analysis, presenting in the "data analysis" section of method the procedure for extracting factors and the criteria for retaining factors. Also, since the correlation between factors is high, it is better to do an oblique rotation.

It makes no sense to include the STAI (line 113) and Rosenberg Self-Esteem Scale (line 109) tests and not to perform any analysis with them. Either they are removed from the method or they are analyzed and the purpose for which they are used is explained.

The discussion should be reviewed once these changes have been made.
